# Mild Crigler–Najjar Syndrome with Progressive Liver Disease—A Multicenter Retrospective Cohort Study

**DOI:** 10.3390/children10091431

**Published:** 2023-08-22

**Authors:** Norman Junge, Hanna Hentschel, Dorothee Krebs-Schmitt, Amelie Stalke, Eva-Doreen Pfister, Björn Hartleben, Martin Claßen, Alexander Querfurt, Veronika Münch, Philip Bufler, Jun Oh, Enke Grabhorn

**Affiliations:** 1Department of Pediatric Kidney, Liver, and Metabolic Diseases, Hannover Medical School, 30626 Hannover, Germany; stalke.amelie@mh-hannover.de (A.S.); pfister.eva-doreen@mh-hannover.de (E.-D.P.); 2Department of Pediatrics, University Medical Center Hamburg-Eppendorf, 20251 Hamburg, Germany; hannaschroeder@hotmail.de (H.H.); d.schmitt@uke.de (D.K.-S.); j.oh@uke.de (J.O.); e.grabhorn@uke.de (E.G.); 3Department of Human Genetics, Hannover Medical School, 30625 Hannover, Germany; 4Institute of Pathology, Hannover Medical School, 30626 Hannover, Germany; hartleben.bjoern@mh-hannover.de; 5Prof.-Hess-Childrens Hospital, Klinikum Bremen Mitte, Gesundheit Nord GmbH, 28205 Bremen, Germany; m.classen@kinderarzt-schacht.de (M.C.); alexander.querfurt@gesundheitnord.de (A.Q.); 6Department of Pediatric Gastroenterology, Nephrology and Metabolic Diseases, Charité Universitätsmedizin Berlin, 10117 Berlin, Germany; veronika.muench@charite.de (V.M.); philip.bufler@charite.de (P.B.)

**Keywords:** liver fibrosis, liver transplantation, Gilbert syndrome, hyperbilirubinemia, UGT1A1, phototherapy, phenobarbital, H39D variant

## Abstract

Crigler–Najjar Syndrome (CNS) with residual activity of UDP-glucuronosyltransferase 1A1 (*UGT1A1*) and no need for daily phototherapy is called mild Crigler–Najjar Syndrome. Most of these patients need medical treatment for enzyme induction (phenobarbital) to lower blood levels of unconjugated bilirubin (UCB). Apart from this, no long-term problems have been described so far. The phenotype of patients with the homozygous pathogenic variant c.115C>G p.(His39Asp) in UGT1A1 is described as variable. Clinical observations of our patients led to the assumption that patients with variant c.115C>G have a mild CNS phenotype while having a high risk of developing progressive liver disease. For mild CNS disease, progressive liver disease has not been described so far. Therefore, we conducted a retrospective multicenter analysis of 14 patients with this particular variant, aiming for better characterization of this variant. We could confirm that patients with variant c.115C>G have a high risk of progressive liver disease (seven of fourteen), which increases with age despite having a very mild CNS phenotype. Earlier predictors and causes for an unfavorable disease course are not detectable, but close follow-up could identify patients with progressive liver disease at the beginning. In conclusion, these patients need close and specialized follow-up. Our study questions whether fibrosis in the CNS is really driven by high amounts of UCB or phototherapy.

## 1. Introduction

Crigler–Najjar syndrome (CNS) is an ultra-rare autosomal recessive disorder caused by several known variants in the UDP-glucuronosyltransferase 1A1 (*UGT1A1*) gene, leading to a lack or absence of activity of its encoded enzyme [1]. UGT1A1 is responsible for the glucuronidation of bilirubin. Therefore, children with this hereditary disease present with severe unconjugated hyperbilirubinemia [2]. Depending on the degree of enzyme dysfunction, the CNS is divided into two types [3]. CNS type 1, the more severe form of CNS, is characterized by a total loss of enzyme activity, causing unconjugated bilirubin (UCB) serum levels between 340 and 850 µmol/L (20–50 mg/dL) without treatment. In CNS type 2, nowadays called mild CNS, the enzyme activity is only impaired, leading to UCB levels between 102 and 340 µmol/L (6–20 mg/dL), and can be increased by phenobarbital (PB) [4]. However, not all patients and genetic variants can be clearly assigned to type 1 or type 2, so patients are classified by the clinical course of their response to PB or need for daily phototherapy (PT).

Children with severe phenotype of the CNS are at high risk for irreversible brain damage (kernicterus) and even death if left untreated [5]. In order to prevent kernicterus, these patients require lifelong intensive and daily PT, but compliance and efficiency of phototherapy e usually decrease with age. Therefore, the only definite treatment for severe CNS disease is liver transplantation (LT) still [6,7]. However, further therapeutic options such as hepatocyte transplantation or gene therapy are still under investigation [8,9,10].

In the past, liver parenchyma was considered structurally and histologically normal in both types of CNS, but there is growing evidence of hepatic parenchymal injury presenting as fibrosis in some patients with severe CNS phenotype [11,12]. The etiopathogenesis of fibrosis remains unknown so far as data describing abnormal liver histology in the CNS are limited. Mitchell et al. recently reported significant fibrosis in a cohort of 22 severe CNS patients with low genetic diversity (Amish and Mennonite populations) [6]. So far, liver fibrosis has been found predominantly in patients with severe phenotypes [7]. For the mild CNS phenotype, liver fibrosis or even cirrhosis is not described except in two case reports [5,12]. Since these patients do not need PT or LT, they become old with their native liver and have therefore more time to develop progressive liver fibrosis due to the CNS, theoretically, but the data do not reflect this. Sun et al. described a Chinese cohort of adult mild CNS patients, of whom 15 had liver biopsies, all without any sign of fibrosis [3]. Therefore, initial cases with a mild CNS phenotype but severe liver disease (LD) piqued our interest, and we initiated a retrospective analysis of similar patients in four German Pediatric Hepatology Centers. The aim of this study was the characterization and comparison of mild CNS patients with the variant NM_000463.3: c.115C>G p.(His39Asp) to gain a better understanding of the unusual clinical course and atypical presentation as well as the etiopathogenesis of CNS-associated liver fibrosis.

## 2. Materials and Methods

### 2.1. Patients

For this retrospective observational study, we evaluated the clinical course of 14 patients with the homozygous pathogenic (class 5) *UGT1A1* variant NM_000463.3: c.115C>G p.(His39Asp) and mild CNS phenotype (no need for phototherapy) from four Pediatric Hepatology Centers in Germany (Hamburg, Hanover, Berlin, and Bremen). Data are based on medical records. All patients from the participating centers with clinical suspicion of CNS from 1990 to 2023 were evaluated for the c.115C>G variant, some retrospectively. All patients with this variant were included in this study. Additionally, family members of the identified patients were screened for this variant. Patients have been extensively examined for other liver diseases or metabolic diseases. In patients with progressive LD, a liver biopsy was performed. We obtained the informed consent of all included patients for genetic analysis and research. The study was performed in accordance with the principles of the Declaration of Helsinki. For data management, we used the pseudonymization procedure, in which personally identifiable information fields within the data record are replaced by a pseudonym under the control of the local PI.

Genetic variants of three patients (patient 6 = LP49; patient 7 = LP7; patient 8 = LP74) are also described in an earlier publication (“Diagnosis of monogenic liver diseases in childhood by next-generation sequencing” [13]).

Since chronic liver diseases develop over time and our cohort is mixed in age, we divided the cohort by age: group 1 ≤ 8, group 2 > 8 years of age.

### 2.2. Biochemical Data

Serum liver enzymes (alanine aminotransferase (ALT), aspartate aminotransferase (AST), and gamma glutamyl transpeptidase (GGT)) have been measured in U/L and bilirubin in µmol/L. Upper limits of normal (ULN) are based on the average ULN for standard techniques in European laboratories.

Biochemical data were collected from the medical record at the time of diagnosis and the latest follow-up, and in the case of LT, before and, if available, after LT.

### 2.3. Liver Histology

Liver fibrosis was evaluated by liver histology from liver biopsies and explanted livers in the case of LT. Histology was evaluated by senior liver pathologists of the involved liver transplant centers (Hamburg, Hannover, and Berlin). The fibrosis stage was specified according to the ISHAK Fibrosis Score [14]. In some patients, liver stiffness measurements (LSM) were available. LSM were performed with the Fibroscan^®^ Mini+430 (Echosens, Paris, France) according to the manufacturer’s instructions with either a small probe (S1/S2) or a medium probe (depending on the patient’s breast circumference).

### 2.4. Genetic Anylysis

The technique of genetic testing has changed over time. In a few patients, Sanger sequencing for UGT1A1 was performed; in others, next-generation panel sequencing (21 gene hepatopathy panel) or whole exome sequencing (WES) (n = 6) was performed to exclude other genetic liver diseases. Panel sequencing, WES, variant interpretation, and classification were performed as described elsewhere [13,15].

## 3. Results

From 1990 until 2023, we identified 14 patients (six females) with a c.115C>G missense variant in the homozygous state in exon 1 of *UGT1A1*, leading to the substitution of histidine to aspartic acid at amino acid position 39 in the UGT1A1 enzyme (p.(His39Asp)). Regarding the CNS (the elevation of UCB), the phenotype of these patients was very mild. However, with increasing age, a notable portion of patients developed severe liver disease (LD). The mean age at diagnosis was 3.67 years (median 2.5 years, SD 3.85 years, 95% CI 1.22–6.11), with a follow-up period of 6 months to 42 years. All patients analyzed (n = 12/14) showed the homozygous *UGT1A1* promotor variant c.-41_-40dup also known as the A(TA)_7_TAA or *UGT1A1**28 allele, which is associated with the Gilbert syndrome. Seven out of fourteen patients had severe liver fibrosis or cirrhosis, leading to endstage LD necessitating LT (n = 6) and/or leading to death (n = 1 without LT, n = 1 after LT). Despite an extensive differential diagnostic workup and detailed genetic analysis, we could not detect any other known LD in these patients. In some patients, benign variants (single nucleotide polymorphisms (SNP)) described to predispose to cholestatic liver disease could be detected in heterozygous status [16,17], e.g., ABCB11 variant c.1772A>G, p.(Asn591Ser) or ABCB11 variant c.1331T>C, p.(Val444Ala). Further details on the patient’s characteristics are shown in Table 1, and if available, the pedigrees are shown in Figure 1.

### 3.1. Patient’s Clinical Courses

Figure 1 gives an overview of most of the patients and their families. Following, we describe the clinical course of each patient in our study cohort.

#### 3.1.1. Patients One, Two, and Three (Figure 1a)

Patient one (a female) was the eight child of unrelated Lebanese parents. She was the cousin of patients two and three. Two of her brothers died at the age of 18 from an unknown liver disease (Figure 1a). She became apparent at the age of 10 years with jaundice and unconjugated hyperbilirubinemia (UCB = 274 µmol/L) and was treated with PB subsequently, leading to a decrease in UCB levels. Transaminase levels were slightly elevated, while bile acids were in the normal range. At that time, a liver biopsy showed no pathological changes except mild signs of cholestasis. Genetic testing revealed the variant in exon 1 (c.115C>G) and in the promotor region of *UGT1A1* c.-41_-40dup, both in the homozygous state. At the age of 15 years, she presented with progressive jaundice, cholestasis, portal hypertension with severe splenomegaly, and thrombocytopenia, while liver synthesis was marginally restricted. At this time, a liver biopsy confirmed biliary cirrhosis with vanishing bile ducts. She died from a severe bleeding event during catheter placement for a MARS procedure (Molecular Adsorbents Recirculating System) in the course of hepatic failure.

Patient two (male) is the child of consanguineous Lebanese parents. He became apparent with jaundice and unconjugated hyperbilirubinemia. He was treated with PB. Genetic testing revealed the c.115C>G variant in UGT1A1 (homozygous). According to family history, two cousins of a non-consanguineous relationship died from an undetermined liver disease at 18 years of age (Figure 1a). Other affected family members are his brother (patient three) and his cousin (patient one). The latest data showed UCB elevation (total bilirubin = 185 µmol/L, UCB = 178 µmol/L), elevated transaminase levels (AST = 100 U/L, ALT = 241 U/L), as well as an elevated GGT of 132 U/L. A liver biopsy showed moderate signs of centrolobular perisinusoidal fibrosis along with remarkable cholestasis. In 2013, he was listed for liver transplantation due to liver cirrhosis but was lost for follow-up.

Patient three (male) is the brother of patient two, the child of consanguineous parents, and became apparent with jaundice and hyperbilirubinemia (Figure 1a). He was treated with PB without needing phototherapy. Genetic testing revealed the described homozygous c.115C>G variant in UGT1A1. He developed advanced hepatopathy, portal hypertension, and esophageal varices and was liver transplanted at the age of 12 years. A liver biopsy of the explanted liver showed signs of portal fibrosis, partly with septa, and additional signs of mild chronic portal and intralobular cholestasis. Nine years after LT, the clinical condition and organ function of this patient were good. Bilirubin levels have completely normalized.

#### 3.1.2. Patient Four

Patient four (a female) was presented at 4 years of age due to unconjugated hyperbilirubinemia first. Genetic testing revealed the described variant (c.115C>G) in exon 1 and in the promotor (c.-41_-40dup) in *UGT1A1*, both in the homozygous state. A liver biopsy at this age showed intralobular cholestasis and portal and intralobular inflammation but no cirrhosis. A follow-up liver biopsy 3 years later (at the age of 7 years) showed advanced liver fibrosis. The patient developed cholestasis and impairment of liver function. Another year later (at the age of 8), the girl was liver transplanted, and the explanted liver showed cirrhosis.

#### 3.1.3. Patient Five

Patient five (male) is a child of consanguineous Lebanese parents and was diagnosed with mild CNS at the age of 18 months due to jaundice. Genetic testing revealed the described variant (C.115C>G) in exon 1 and in the promotor of *UGT1A1* (c.-41_-40dup). Laboratory values showed UCB elevation (total bilirubin = 140 µmol/L, UCB = 133 µmol/L) and slightly elevated transaminase levels (AST = 37 U/L, ALT = 86 U/L). At this point, a liver biopsy showed no sign of cholestasis or fibrosis. At the age of 13 years, the patient was suspected to have primary sclerosing cholangitis based on magnetic resonance cholangiopancreatography findings. A liver biopsy revealed cholestasis without fibrosis. The latest data at the age of 16 years showed elevated transaminase levels (AST = 71 U/L, ALT = 181 U/L), while total bilirubin and UCB were 322 µmol/L and 308 µmol/L, respectively. The boy was in good general health. After this, he was lost for follow-up.

#### 3.1.4. Patients Six and Seven (Figure 1b)

Patient six (male) is the child of consanguineous Turkish parents and the cousin of patient seven (Figure 1b). At the age of six months, he was referred to a liver center due to UCB elevation (total bilirubin = 153 µmol/L, UCB = 137 µmol/L), elevated transaminase levels (AST = 116 U/L, ALT = 181 U/L), and due to family history (see patient 7). Genetic testing revealed the c.115C>G variant and the c.-41_-40dup variant in *UGT1A1*, both in the homozygous state. Therapy was maintained through PB and ursodeoxycholic acid (UDCA). He was already treated with PB at his first presentation; therefore, we cannot be sure about how his UCB levels would have been without it. The latest data at the age of nine showed slightly elevated AST and ALT and mild UCB elevation under low-dose PB and no signs of liver fibrosis based on Fibroscan^®^ (Echosens, Paris, France) results (4.1 kPa).

Patient seven (female) was the child of non-consanguineous Turkish parents suffering from genetically confirmed mild CNS (homozygous c.115C>G in *UGT1A1*) and suspected GS due to the homozygous c.-41_-40dup promotor variant. She became symptomatic at the age of five years with jaundice and UCB elevation (UCB = 198 µmol/L, conjugated bilirubin = 2 µmol/L), elevated transaminase levels (AST = 134 U/L, ALT = 201 U/L), as well as elevated cholestasis parameters (alkaline phosphatase (AP) = 438 U/L, GGT = 438 U/L). At this point, liver biopsy showed focal portal fibrosis (ISHAK 1) and minor canalicular cholestasis (Figure 2a). Initially, she was treated with PB, which lowered UCB serum levels. UDCA was added later. After recurrent episodes of gallstone-related cholecystitis, she underwent a cholecystectomy. During the following years, her condition worsened, and she finally came down with acute-on-chronic hepatic failure due to not classifiable chronic cholestatic liver disease at the age of almost 16 years. At that time, nine years after her first liver biopsy, she presented with histologic signs of severe canalicular and ductular cholestasis along with severe biliary fibrosis (Figure 2b). Clinically, she developed increasing ascites and rectal bleeding, while ultrasound showed a retrograde flow of the portal vein due to cirrhosis. Her further course was complicated by hepatic encephalopathy and hepatorenal syndrome. She underwent LT at 16 years of age. The explanted liver showed biliary cirrhosis with enormous cholestasis (Figure 2c). After persistent respiratory impairment and cardiopulmonary resuscitation, she unfortunately died four weeks after transplantation of hypoxia-induced severe cerebral edema.

#### 3.1.5. Patient Eight (Figure 1c)

Patient eight (male) is the child of likely consanguineous Turkish parents and presented at the age of 3 years with jaundice and UCB elevation (total bilirubin = 139 µmol/L, UCB = 121 µmol/L), elevated transaminase levels (AST = 88 U/L, ALT = 161 U/L), as well as mildly elevated GGT (91 U/L). He had a sister with mild CNS who died of liver failure at the age of 18 (Figure 1c). His genetic testing revealed the c.115C>G variant and the promotor variant c.-41_-40dup in *UGT1A1*, both in the homozygous state. A liver biopsy showed signs of mild cholestasis and fibrosis (ISHAK Fibrosis Score 2), liver cell swelling, and single liver cell death. He was treated with PB and UDCA. The latest data from 2018 (age 7.5 years) showed normalized transaminase levels while bilirubin levels remained stable (total bilirubin = 145 µmol/L, UCB = 140 µmol/L).

#### 3.1.6. Patients Nine and Ten (Figure 1d)

Patient nine (male) is the child of consanguineous Lebanese parents and the older brother of patient ten. He had a further brother who died at the age of 11 from liver cirrhosis (Figure 1d). Genetic testing revealed homozygosity for the exon 1 mutation of *UGT1A1* (c.115C>G) and the promotor variant c.-41_-40dup. At the age of 13 years, this boy presented with jaundice, pruritus, and varices. Laboratory values showed UCB elevation (total bilirubin = 144 µmol/L, UCB = 132 µmol/L), elevated transaminase levels (AST = 89 U/L, ALT = 105 U/L), as well as elevated cholestasis parameters. A liver biopsy revealed severe fibrosis (ISHAK Fibrosis Score 5) and cholestasis. One year later, LT was performed in this patient, followed by re-LT 8 days later due to hyper-acute cellular rejection of the transplanted liver. Histology from the explanted native liver showed liver cirrhosis with cholestasis and bile plugs. Currently, he is 43 years old, and his condition is stable with respect to the liver but complicated by cardiac disease caused by aortic valve insufficiency.

Patient ten (female) is the child of consanguineous Lebanese parents and the sister of patient nine (Figure 1d). Genetic testing for *UGT1A1* revealed the same results as for her brother. At the age of five years, she presented with mild unconjugated hyperbilirubinemia (total bilirubin = 60 µmol/L, UCB = 56 µmol/L) and slightly elevated transaminase levels (AST = 39 U/L, ALT = 73 U/L). At the age of 15, her condition worsened. Laboratory values showed signs of progressive cholestasis and reduced liver synthesis. Furthermore, she presented with significant splenomegaly. LT was performed two months later; histology of the explanted liver revealed cirrhosis (ISHAK 5) and cholestasis. Twenty days after LT, she underwent re-LT due to arteria hepatica thrombosis. The latest biochemical data from March 2021 showed normal liver enzymes, no sign of cholestasis, and good liver transplant function. At this time, the patient was 35 years old and in good clinical condition.

#### 3.1.7. Patient 11

Patient 11 (female) is a 12 month old girl of consanguineous parents from Turkey. She presented at the age of 3 months with a high UCB (476 µmol/L). Genetic testing revealed homozygosity for the exon 1 variant of *UGT1A1* (c.115C>G) and the promotor variant c.-41_-40dup. Phototherapy was started first, and after stabilizing the UCB, PB was added and PT was reduced. Due to the increase in GGT, the PB withdrawn, and the UCB stayed stable at 85 µmol/L. Up until now, the patient has been in good condition without treatment and without signs of progressive liver disease.

#### 3.1.8. Patient 12,13,14

Patient 12 (female) is a 5 year old girl from a consanguineous Lebanese family and a cousin of the siblings, patients 13 and 14. She was diagnosed with CNS at the age of three. She has the typical genetic constellation for *UGT1A1* (homozygosity for c.115C>G and the promotor variant c.-41_-40dupA). At diagnosis, she had elevated AST, ALT, and UCB (143 µmol/L) but no cholestasis. However, a liver biopsy at the age of one year showed minimal signs of cholestasis but no fibrosis. She was treated with PB for a short period. At the last follow-up, UCB was stable (135 µmol/L) without any treatment, and AST/ALT were still increasing discreetly but without signs of liver fibrosis so far.

Patient 13 (male) is a one year old boy, who was diagnosed by genetic testing within 42 days after birth due to the diagnosis of his brother (Patient 14). Both show the typical genetic constellation for *UGT1A1* (homozygosity for the c.115C>G and the promotor variant c.-41_-40dup). UCB was 162 µmol/L at this time. His parents are from Lebanon. The last UCB was 46 µmol/L without treatment.

Patient 14 (the male, older brother of patient 13) is 3 years old and was diagnosed at one year of age. UCB was 125 µmol/L at this time. Both patients (13 and 14) had PT courses after birth but no PB. The last UCB was 100 µmol/L without any treatment. Both patients showed elevated AST and ALT at diagnosis and still (P13: AST 106 U/L, ALT 144 U/L; P14: AST 178 U/L, ALT 291 U/L). Patient 14 had an elevated GGT and conjugated bilirubin (18.5 µmol/L) at the last follow-up. However, both show no clinical signs of progressive liver disease so far (normal liver function, no splenomegaly).

### 3.2. Liver Enzymes and Function According to the Patient’s Age

The patients were split into two age groups. Group one up to the age of 8 years (n = 6) and group two (n = 8) starting with the age of 9 years. We found higher values for AST, ALT, GGT, total bilirubin, and INR in the older group. Most pronounced was this difference between the groups for ALT (mean 127.40; median 105.00 versus 162.00; 143.00 U/L; *p* = 0.44), GGT (146.13; 129.8 versus 236.75; 179.00 U/L; *p* = 0.13), INR (1.20; 0.94 versus 1.51; 1.08; *p* = 0.08), and total bilirubin (146.13; 48.00 versus 236.75; 179.00 µmol/L; *p* = 0.11) but not for conjugated bilirubin.

### 3.3. Liver Enzymes and Function According to the Need for Liver Transplantation

The following parameters from the date of diagnosis were tested for their association with the subsequent need for LT: AST, ALT, GGT, total bilirubin, conjugated bilirubin, and INR. For none of them, an association could be shown.

The same paramters, but from the date of the last visit at follow-up or the last visit before LT, showed higher values in patients with a need for LT (Table 2). For GGT, this difference was significant (*p* = 0.03). The interval from last visit before LT to LT was 0–44 weeks (mean 20.40, median 13.00 weeks) in these patients.

## 4. Discussion

Studies about liver fibrosis or even cirrhosis in the mild CNS phenotype (former type 2) do not exist so far. Only two case reports exist. One of them describes a patient with the same genetic variant as our 14 patients (c.115C>G variant leading to the H39D enzyme variant of the UGT1A1 enzyme). Initially, this variant was thought to lead to a severe phenotype [18,19], but already others have reported a mild phenotype of the CNS with this variant [5,20]. This is particularly evident in combination with the c.-41_-40dup variant of the *UGT1A1* promoter [20]. Maruo et al. [20] showed by in vitro enzyme activity measurement that the H39D variant has residual activity between 4.2 and 8.1% of normal UGT1A1 activity at physiological pH. Our patients presented with mild elevations of UCB but were far from needing phototherapy to prevent kernicterus. Only four patients received low-dose PB treatment for a longer or ongoing period to lower their UCB levels. All patients with progressive LD showed biliary cirrhosis with aspects of cholestasis in the liver histology but without clear signs of cholestasis in the biochemical profile initially. Therefore, we could not find any association between blood values at diagnosis and the subsequent need for LT.

Nine out of fourteen patients developed some kind of fibrosis, and seven out of fourteen had even severe fibrosis, leading to liver failure necessitating LT or leading to death. Notably, most of them had normal liver histology or absent signs of liver disease at an early age but developed both in adolescents. We could clearly show that in this cohort, the biochemical signs of LD are increasing with age and are higher in patients over the age of 8 years. We could show that patients with a need for LT developed higher values for AST, ALT, and INR and significantly higher ones for GGT during the course of the disease. This means that patients with suspicion for advanced LD can be identified by their biochemical profile, but still, an overall clinical impression is important. That GGT shows a significant difference underlines the cholestatic picture of this unclear chronic LD, even though direct/conjugated bilirubin is not higher in patients with need for LT compared to them without. Unfortunately, we have serum bile acids for only half of the patients at this point in time. Only a few patients received medical treatment. Four patients received PB at least for a while, and four received UDCA at an older age. We could not analyze the association of the treatment with the outcome since UDCA was only given to patients with progressive LD. In the future, it would be interesting to investigate whether an early start of UDCA (before the onset of progressive LD) could improve outcomes. In summary, our results underline the importance of follow-up in a specialized liver center on a 6 or 12 monthly routine for these patients to detect signs of progressive LD early.

The pathogenesis of progressing LD in children with a mild CNS phenotype due to the homozygous c.115C>G variant is still unresolved and cannot be thoroughly explained by a pathophysiological understanding of the CNS. Not even the pathophysiology of fibrosis in severe CNS, a phenomenon that is described and investigated much more, is clearly understood so far [7]. Different possible causes are discussed, for example, biliary obstruction secondary to cholelithiasis, toxicity related to heme degradation products, or bilirubin photoisomerization products [7]. The latest cannot be the cause in our patients since they have not been treated with phototherapy. The first aspect, cholestasis, is described as unlikely in severe CNS since analysis of the CNS registry could not detect any correlation between direct bilirubin or GGT and fibrosis [7]. However, in this cohort, we found an association between GGT and the need for LT. Therefore, subclinical cholestasis may play a role. It is unclear if this cholestasis and progressive LD are caused by the CNS or other additional gene variants (maybe a heterozygous one). A genetic cause seems likely since we saw strong intra-family accumulation. Additionally, the medical history of patients in our study cohort detected at least five unclear deaths in childhood, possibly associated with liver disease. Therefore, trio analyses as well as multi-sample analyses were performed for patient six. We especially focused on variants in proximity to/on the same chromosome as the UGT1A1 variant, as we expected a genetic linkage between the UGT1A1 variant and a variant potentially responsible for the progressive LD. However, we could not identify any candidate genes. The detected heterozygous benign genetic variants (SNP) described to predispose for cholestatic liver disease are frequent in the normal population and therefore may favor but cannot be the only cause of liver cirrhosis in these patients, especially as these variants were detected only in some of our patients. Possibly, the *UGT1A1* variant c.115C>G itself does not only lead to CNS but also has a pathomechanism different from all other known pathogenic UGT1A1 variants, leading to progressive LD.

## 5. Conclusions

In conclusion, we suggest a careful clinical follow-up of patients with a mild CNS phenotype (especially patients with the UGT1A1 c.115C>G variant) for progressive LD since it can develop in adolescents or young adults. In cases of progressive LD, patients should be considered for LT. Liver fibrogenesis in the CNS might rather not be associated with severity of the CNS-phenotype (height of UCB serum values) but with additional pathomechanisms of certain UGT1A1 variants or other gene variants outside UGT1A1. Functional studies are needed to improve the pathophysiological understanding of liver fibrosis in the CNS.

## Figures and Tables

**Figure 1 children-10-01431-f001:**
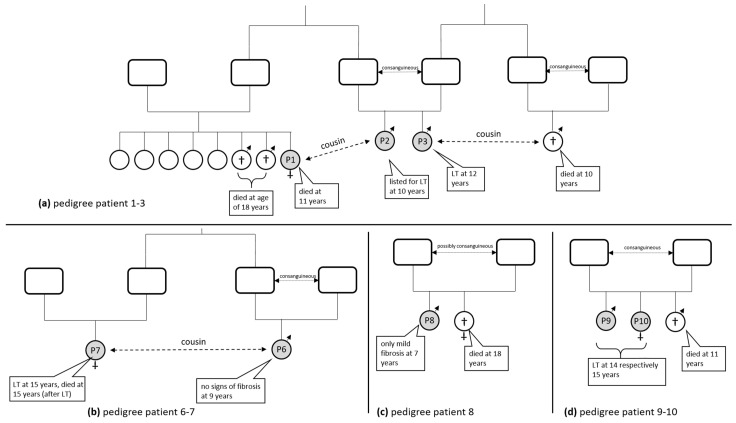
Pedigrees of the families (**a**) from patients 1–3; (**b**) from patients 6 and 7; (**c**) from patient 8; and (**d**) from patients 9 and 10, including relatives reported in the family history who died due to unknown liver disease. LT = liver transplantation.

**Figure 2 children-10-01431-f002:**
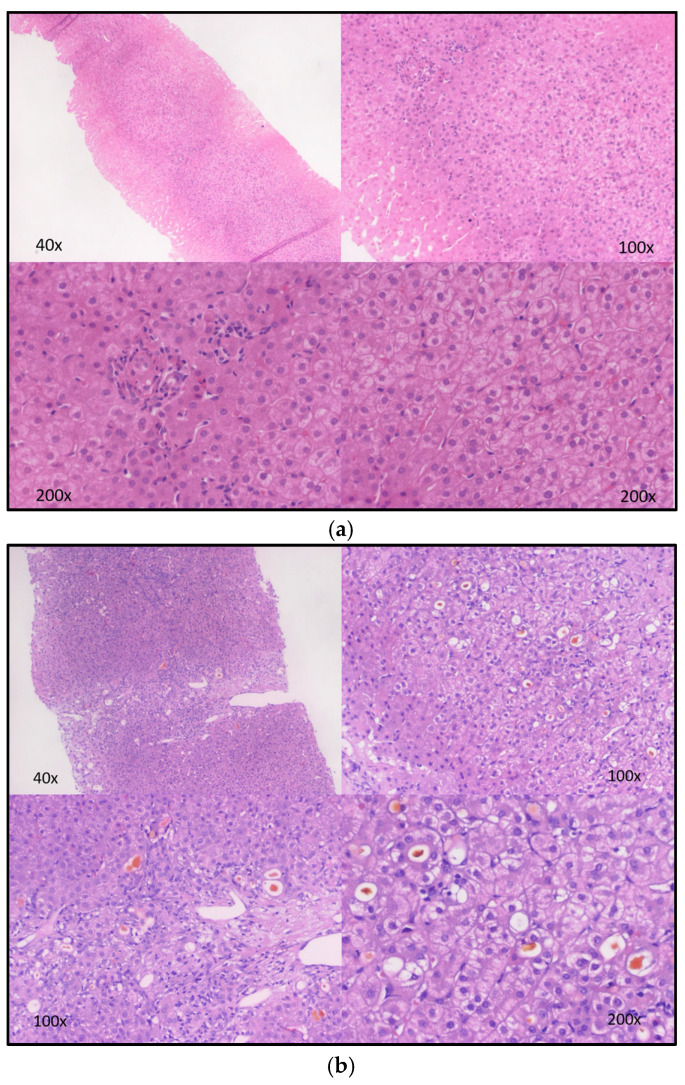
Histology images from liver biopsies and the explanted liver of patient seven. (**a**) The first liver biopsy at the age of 5 years shows portal inflammation, some focal portal fibrosis, and minor canalicular cholestasis (not seen on the images). (**b**) A second liver biopsy at the age of 14 years (8 months before LT) shows bridging fibrosis, ductular reaction, and severe cholestasis. (**c**) Shows the histology of the patient’s explanted liver during LT with cirrhosis, ductular reaction, and severe cholestasis.

**Table 1 children-10-01431-t001:** Patient characteristics.

Variable (n = Available Patients/Data)	Median (Mean); or Characteristic and Corresponding Absolute Number	95% CI; or Percentage of Available Patients/Data	Minimal– Maximal Value
Sex (n = 14)	female n = 6	43%	---
Patients with promotor variants (n = 12)	n = 12	100%	---
Treatment with phenobarbital	n = 4	29%	---
Patients with LT	n = 6	43%	---
Patients alive	n = 12	86%	
Patients with advanced fibrosis (ISHAK F > 3) (n = 11)	n = 7	64%	
Age at diagnosis (n = 8)	2.50 (3.67) years	1.22–6.11	0.08–12.00
Age of the last liver biopsy before LT	12.00 (10.80) years	6.96–14.64	1.00–17.00
Age at liver biopsy: non-advanced fibrosis (ISHAK F ≤ 3)	12.00 (9.00) years	0.62–17.38	1.00–17.00
Age at liver biopsy: advanced fibrosis (ISHAK F > 3)	14.00 (12.60) years	8.43–16.77	7.00–15.00
Age LT (n = 5)	12.40 (13.00) years	8.32–16.48	7.00–15.00
AST before LT or last visit (n = 13)	105.00 (146.15) U/L	63.71–228.60	28.00–545.00
ALT before LT or last visit (n = 12)	124.5 (152.33) U/L	89.05–215.61	45.00–318.00
GGT before LT or last visit (n = 12)	54.00 (103.75) U/L	3.02–204.48	18.00–584.00
Total bilirubin before LT or last visit (n = 13)	146.00 (202.14) µmol/L	119.85–284.43	55.00–555.00
Conjugated bilirubin before LT or last visit (n = 12)	9.5 (39.44) µmol/L	−05.39–84.28	4.00–214.00
Bile acids before LT or last visit (n = 6)	229.33 (179.00) µmol/L	−30.44–489.11	7.00–570.00
Albumin before LT or last visit (n = 12)	40.4 (39.68) g/L	35.93–43.4	29.00–48.00
INR before LT or last visit (n = 11)	1.05 (1.37)	0.92–1.82	0.80–2.46
Thrombocytes before LT or last visit (n = 13)	226.00 (220.77) Tsd/µL	155.76–285.78	68.00–403.00
Interval diagnosis—LT (n = 4)	5.8 (5.77) years	−3.83–15.37	0.25–11.16
Age at death (n = 2)	15.00 (15.00) years	15.00–15.00	15.00–15.00
Patient age at last follow-up (n = 14)	12.00 (13.64) years	6.50–20.79	0.60–42.00
Patient age at last follow-up (n = 14)	148.00 (170.00) months	84.56–255.44	6.00–506.00
Patient age at last follow-up only w/o need for liver transplantation (n = 8)	6.00 (6.75) years	1.51–11.99	0.60–16.00

LT = liver transplantation, AST = aspartate aminotransferase, ALT = alanine aminotransferase, GGT = gamma-glutamyl transferase, INR = international normalized ratio, w/o = without.

**Table 2 children-10-01431-t002:** Variables at the last visit and need for liver transplantation.

Variable	Patients with Need for LT (n = 6) Median (Mean) 95% CI	Patients without Need for LT (n = 8) Median (Mean) 95% CI	*p* =
Age at blood analysis (last visit or last visit before LT)	13.5 (13.00) 8.96–17.04	6.00 (6.50) 1.31–11.69	0.08
AST (U/L)	104.00 (179.17) −9.54–367.87	88.50 (114.25) 43.39–185.11	0.49
ALT (U/L)	173.00 (167.67) 47.16–288.17	105.00 (132.43) 55.75–209.11	0.73
GGT (U/L)	82.00 (184.60) −94.87–464.07	28.5 (50.50) 7.00–93.99	0.03
Total bilirubin (µmol/L)	179.00 (203.67) 128.48–278.86	129.8 (193.60) 52.37–334.83	0.23
Conjugated bilirubin (µmol/L)	9.20 (48.60) −66.28–163.48	9.51 (29.04) −16.37–74.45	0.94
INR	1.71 (1.73) 0.53–2.92	0.99 (1.17) 0.64–1.70	0.73

LT = liver transplantation, AST = aspartate aminotransferase, ALT = alanine aminotransferase, GGT = gamma-glutamyl transferase, INR = international normalized ratio.

## Data Availability

Due to the nature of this research, participants in this study did not agree to their data being shared publicly, so supporting data are not available.

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
