# Peer review of "Mild Crigler–Najjar Syndrome with Progressive Liver Disease—A Multicenter Retrospective Cohort Study"

_children, 2023, doi:10.3390/children10091431_

Round 1

Reviewer 1 Report

The text should have a correct way of numbering the paragraphs: at chapter 3.1 the subsequents numbers are always 1.1.1, for each group of the cases presented.

It could be easier to understand and follow the results if the units for bilirubin would be the same. (in some places it is measured in mg/dl in other ones in µmol/l )

no comments necessary

Author Response

We thank the reviewer for the thorough review of our manuscript and the very helpfull comments. We harmonized the manuscript regarding the unit for bilirubin levels and we corrected the numbering of the paragraphs.

Reviewer 2 Report

authors conducted an interesting research about the Crigler Najjar Syndrome (CNS), comparing mild CNS patients with the variant NM_000463.3: c.115C>G p.(His39Asp) to gain a better understanding of the unusual clinical course and atypical presentation and the CNS-associated liver fibrosis.

Here are some small corrections:

-        Line 46: close brackets

-        Line 94 put “:” instead of “;”

-        line 176: remove “first”, it is is a repeat

-        line 193 in the phrase "UCB were 18.8 mg/dl and 18.0 mg/dl” change the unit of measurement to µmol/l

-        line 213 use PB instead of Phenobarbital

-        Lines 260 -261: Reformulate the Sentence as follows: At this time, the patient is 35 years old and in good clinical condition.

-        Fig 1: P2 E P3: write "LT at 10 years" and "LT at 12 years"; P9 and P10: "LT" instead of "liver transplant"

-        Line 144: Instead of "Liver Transplantation (LT)" enter only "LT", acronym already specified

-        Line 364: Instead of "liver disease (LD)" enter only "LD", acronym already specified

-        Line 374; "patients" instead of "patient"

-        Lines 414-415: replace "since LD can develop...." With "Since it can develop..."

Good

Author Response

We thank the reviewer very much for the detailed review of our manuscript and the very helpful comments. Following this comments we could further improve the manuscript. We implemented all suggestions, except the suggested change for Patient #2 in Fig1. Change text to "LT at 10 years" from "with 10 years need for LT" would be not correct, since the patient was listed for LT but then lost for follow up. We changed to "listed for LT at 10 years". We hope the reviewer agree to this.

Reviewer 3 Report

An interesting case series demonstrating the presence of progressive liver disease in a significant proportion of patients with mild CNS.

Minor language and formatting comments only:

1. in abstract line 22-23: "...in UGT1A1 is described variable." should read "...in UGT1A1 is described as variable."

2. in the introduction line 45-47 " In CNS type 2, nowadays called mild CNS, the enzyme activity is only impaired leading to UCB levels between 6 and 20 mg/dl (102-340µmol/l and can be increased by phenobarbital"  there is a bracket missing after (102-340µmol/l.  Might also be clearer as follows " In CNS type 2, nowadays called mild CNS, the enzyme activity is only impaired, leading to UCB levels between 6 and 20 mg/dl (102-340µmol/l), and can be increased by phenobarbital"

2. Table 1 should not be split over two pages because the headings for each column cannot be seen on the second page

3. In the legend for Figure 1 - "anamnestic" relates only to immunological memory and not to cognitive memory recall.

4. Again in Figure 1 several of the labels don't make sense: "with 10 years need for LT", "LT with 12 years", "LT with 15 years, death with 15 years", "no signs of fibrosis with 9 years" and "only mild fibrosis with 7 years" should all instances of "with" be replaced by "at"?

5. Reference 20 has all the authors in capital letters!

generally very good, minor comments on language see above

Author Response

We thank the reviewer very much for the precise and very helpful review of the manuscript, which helped to further improve the manuscript. We changed our manuscript based on the recommendations.